CellPress

## Perspective

# Galaxy single-cell & spatial omics community update: Navigating new frontiers in 2025

Marisa Loach,[1,12] Amirhossein Naghsh Nilchi,[2,3,12] Diana Chiang,[2,12] Morgan Howells,[6,12] Florian Heyl,[4] Helena Rasche,[9] Julia Jakiela,[8] Mehmet Tekman,[3] Menna Gamal,[10] Pablo Moreno,[5] Saskia Hiltemann,[7] Timon Schlegel,[2] Björn Grüning,[2] Rolf Backofen,[2,11] Pavankumar Videm,[2,13,*] and Wendi Bacon[1,13,*]

[1]School of Life, Health and Chemical Sciences, The Open University, Milton Keynes, UK
[2]Department of Computer Science, University of Freiburg, Freiburg, Germany
[3]Institute of Experimental and Clinical Pharmacology and Toxicology, Faculty of Medicine, University of Freiburg, Freiburg, Germany
[4]German Human Genome-Phenome Archive (GHGA) and the Division of Computational Genomics and Systems Genetics, German Cancer Research Center, Heidelberg, Germany
[5]Oncology Data Science & AI, Oncology R&D, AstraZeneca, Cambridge, UK
[6]School of Computing and Communications, The Open University, Milton Keynes, UK
[7]Central Data Facility, University of Freiburg, Freiburg, Germany
[8]School of Chemistry, University of Edinburgh, Edinburgh, UK
[9]Department of Clinical Bioinformatics, Erasmus Medical Center, Rotterdam, the Netherlands
[10]Bioinformatics Department, University of Birmingham, Birmingham, UK
[11]Signalling Research Centre CIBSS, University of Freiburg, Freiburg, Germany
[12]These authors contributed equally
[13]These authors contributed equally
*Correspondence: videmp@informatik.uni-freiburg.de (P.V.), wendi.bacon@hdruk.ac.uk (W.B.)

## SUMMARY

Single-cell omics, named Method of the Year three times, have revolutionized biological research by enabling the high-resolution exploration of cellular heterogeneity and molecular processes. Initially centered on transcriptomics, this rapidly evolving field now ranges from multiomics to spatial analysis, with expanding customization options. The ubiquity of such analyses and the lack of a unified pipeline necessitate the development of scalable, flexible, and integrated tools and workflows. The Galaxy platform has responded to these technological advancements, extending its repertoire of freely accessible tools and workflows, backed by expert-reviewed and user-informed training resources to empower researchers to perform and interpret their own analyses. With more than 175 tools, 120 training resources, and 300,000 jobs running at the time of writing, this process has culminated in the development of Galaxy single-cell and spatial omics community (SPOC), designed to promote global collaboration in advancing usable, reproducible, accessible, and sustainable single-cell and spatial omics research.

## INTRODUCTION

Single-cell and spatial omics (SPO) techniques have revolutionized biological research by enabling the exploration of cellular heterogeneity at high resolution, across molecular layers, and *in situ*.[1]

### Evolving technology

Single-cell analysis originated as a means to explore transcript expression at the level of an individual cell—a significant advance from the previous bulk transcriptomics technology that homogenized samples. However, single-cell datasets were often sparse, technically noisy, and often biased.[2] New tools proliferated rapidly as bulk transcriptomics approaches were adapted to address these challenges. As single-cell technologies scaled from profiling tens to thousands of cells, the resulting datasets increased substantially in size and complexity.[3] Public data repositories and cell atlases were built to enable dataset integration across studies, increasing sample size and statistical power.[4]

In addition to size, single-cell datasets have also grown in complexity, as emerging omics technologies now capture multiple molecular modalities within the same cells. Single-cell transcriptomics, single-cell multimodal omics, and spatially resolved transcriptomics were named *Nature Methods* Method of the Year in 2013,[5] 2019,[6] and 2020,[7] respectively. Each emerging technology introduces the need for novel analytical methods, increasingly complex data structures, and greater computational resources.

### Software challenges

The proliferation of software tools has introduced challenges such as incompatible data formats, unstable tools, and the

absence of standardized workflows. Moreover, SPO analyses require iterative exploration and customization, wherein users make complex and subjective decisions throughout the analytical process.[8] Good research data management enables analysis, sharing, and reuse of single-cell data. While the FAIR (findable, accessible, interoperable, and reusable) data principles offer guidance on how to make data findable, accessible, interoperable, and reusable,[9] researchers often need support to apply these principles in practice. End-to-end workflows provide a starting point, but must be supplemented by clear and accessible training materials to guide users in making informed analysis decisions.

### Galaxy for single-cell analysis

Galaxy is a free, open-source, web-based platform for programming-free bioinformatics analysis in a graphical user interface with embedded FAIR features.[10,11] Galaxy combines the reproducibility of other workflow management systems like Nextflow[12] with the user-friendly interfaces of web applications such as Vitessce[13] and CellxGene-VIP (visualization in plugin).[14] The SPO tools now available on Galaxy go beyond the data exploration and visualization provided by these web apps, both of which are incorporated into the platform as interactive tools.

To support single-cell data analysis, the Galaxy community initially built single-cell RNA sequencing (scRNA-seq) analysis portals with tools and workflows. This infrastructure was accompanied by training resources to facilitate its effective use.[15,16] These Galaxy SPO resources have evolved alongside SPO technology to support complex data types, modalities, and increasingly large datasets (Figure 1A). The Galaxy training network (GTN) infrastructure[17] has similarly progressed, enabling notebook-based tutorials, workflow testing, and learning pathways which have facilitated the expansion of the Galaxy SPO training offering (Figure 1D). Following initial publications of Galaxy single-cell tools in 2020[15,16], more than 50 new SPO tools have been added. Galaxy contributors (see Figure 1F) have since published 31 new tutorials, as well as supporting materials including FAQs, video walkthroughs, and slide decks. Each tutorial includes workflows primarily designed to guide users, but which can often be easily modified to suit user data.

To bring together people working on SPO analysis across the world, the Galaxy Single-cell and Spatial Omics Community (SPOC) was formed in 2022. This community aims to enable best practices for FAIR analysis of SPO data by scientists worldwide. This manuscript describes the current SPO offering in Galaxy.

## OVERVIEW OF SPOC

### SPOC: Galaxy single-cell and spatial omics community

The Galaxy single-cell and spatial omics community (SPOC) was launched in 2022 to centralize and streamline community contributions, prevent work duplication, and reduce barriers to engaging with tool and tutorial development. Our Community of Practice page (Table 1, row 1) sets out our goals, publicizes our meetings, and welcomes new members. We established and documented best practices for building communities in Galaxy, which we shared via a new training topic on community

building in the GTN (Table 1, rows 2–4). SPOC contributed to the creation of the Galaxy Community Board in 2023, a much-needed arm of the Galaxy Governance structure (Table 1, row 5) to champion user voice in development.

### Galaxy subdomain for serving community needs

Subdomains[11,20] are curated Galaxy instances hosted on shared servers (.eu, .org, .org.au, and .fr) and aimed at specific communities of users. Before SPOC was created to coordinate contributions, two separate single-cell subdomains were launched on Galaxy, creating confusion among users (Table 1, rows 6 and 7).[15,16] SPOC unified these into a single subdomain where all relevant tools and content have been collected and displayed according to user needs (Table 1, row 8). We signpost this subdomain on all SPO training material to spur adoption (Table 1, row 9). Indeed, our community advocacy on the importance of centralizing and streamlining user access to the resources they need helped drive the uptake of the Galaxy Labs Engine[21] for building centralized subdomains. The three main public Single-cell Galaxy subdomains are accessible through their respective servers: the USA subdomain at https://singlecell.usegalaxy.org, the European subdomain at https://singlecell.usegalaxy.eu, and the Australian subdomain at https://singlecell.usegalaxy.org.au.

### Sustainability

Maintaining the existing tools is a core responsibility for SPOC, in order to avoid software collapse (where tools become unusable because they are not kept in working order[22]). Sustainability in bioinformatics is a notoriously growing problem, where scientists often create bespoke code on short-term contracts and then abandon it for other roles.[23,24] Outdated or failing software is particularly problematic for new users, who are often unable to distinguish user error from tool error. SPOC prioritizes sustainability by coordinating both tool and training material maintenance.

#### *Tools*

To demonstrate our continuous tool development, we extracted the number of commits from the four most popular Galaxy tool repositories containing SPO tools. As shown in Figure 1B, the IUC (Intergalactic Utilities Commission, which enforces tool testing and high-quality tool development), tool repository contains the most SPO tools, followed by the EBI (European Bioinformatics Institute) tool repository. The number of commits reflects a similar trend, with a total of 1,400 commits across all main tool repositories for maintaining 187 tools. This commitment to sustainability results in the consistent use of our Scanpy and Anndata tool suites over time (Figure 2D).

We aim to update tools, integrate new functions, and address user issues. This philosophy can be seen in the recent development of a new Seurat toolsuite on Galaxy (Table 2, row 1). While our main motivation was to ensure Galaxy users could take advantage of the updated features in Seurat version 5,[25] the opportunity was also taken to reorganize the functions to make them easier to find and use, based on training course feedback. The tool suite now includes seven Galaxy Seurat tools, each offering a selection of functions for creating Seurat objects, preprocessing, dimensional reduction, clustering, integration, visualization, and data management. The new tools include options that were

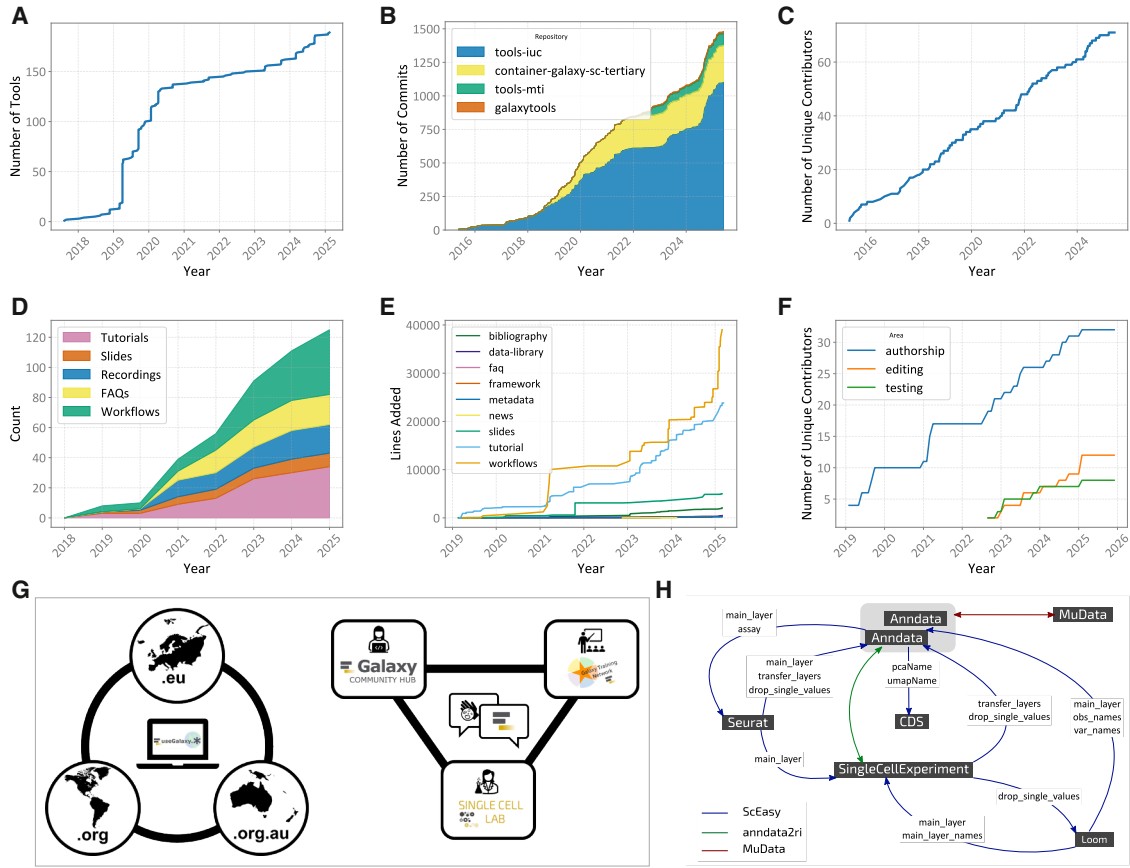

**Figure 1. Galaxy SPO resources and community growth over time**

(A) SPO tools added to the Galaxy platform. Complex tool suites are often divided into multiple Galaxy tools; for instance, the Anndata Python package is split into four distinct tools, each serving a specific purpose—importing, exporting, inspecting, and manipulating Anndata objects. Since 2020, we have gathered more functionality in individual tools, rather than one function per tool, to make tool finding easier for users.

(B) Number of commits in Galaxy SPO tool repositories. Four of the main tool repositories with a total of over 1,400 commits show tool sustainability.

(C) The number of contributors to Galaxy SPO tool repositories steadily increased over time.

(D) SPO training resources added to the GTN. Since 2020, a wide range of new tutorials (31), workflows (38), FAQs (20), recordings (18), and slide decks (7) have expanded the GTN.

(E) Cumulative lines added to SPO training resources within the GTN.

(F) In February 2022, the GTN added support for recording different aspects of a contributor's contributions, which is reflected in the increase in editors and testers for single-cell materials.

(G) Regional, technical, and pragmatic collaboration for SPOC. Left side: collaboration across the original "big three" servers of AU, USA, and EU, which enables shared resources for further federated servers. Right side: collaboration across technical skillsets (user, developer, and trainer) and existing community sites (Galaxy Community Hub; Subdomains/Galaxy Labs; Galaxy Training Network).

(H) The allowed conversions between common SPO data types in Galaxy.

previously unavailable in Galaxy, including completing all preprocessing with the single SCTransform function[26] and a selection of methods for performing batch correction or dataset integration. A new tutorial was also created (Table 1, row 10).

### Training

Our commitment to sustainability drove the creation of a Maintainer page (Table 1, rows 11 and 12) to monitor best practices, updates, and multi-server testing of training workflows. The coordination of updates by SPOC and biannual training events has driven user-powered updates. In the last three years, our focus has shifted from making single large commits infrequently to making smaller changes more often (Figure 1E).

### Workflows

We further repurposed the rigid workflows used in our tutorials into more flexible, generic workflows. The workflows from our previously published scRNA-seq preprocessing tutorials are now featured on Dockstore[38] (Table 1, rows 13 and 14). Additionally, the workflow from our most popular tutorial on clustering peripheral blood mononuclear cells (PBMCs) with Scanpy was adapted to work with any Anndata object, allowing users to customize key parameters for clustering and plotting (Table 1, row 15). Developed as a part of the Intergalactic Workflow Commission repository,[39] these workflows are equipped with robust tests and receive automatic updates whenever the underlying tools are updated, ensuring long-term sustainability.

**Table 1. Resources and events for SPOC, subdomains, sustainability, and engaging users**

| Item | Type | Subsection | Description |
|------|------|------------|-------------|
| 1 | ♣ | SPOC | Site: SPOC Community Hub |
| 2 | 🖥 | SPOC | What's a Special Interest Group? |
| 3 | 🖥 | SPOC | Creating a Special Interest Group |
| 4 | 🖥 | SPOC | Creating Community Content |
| 5 | ♣ | SPOC | Site: Galaxy Community Board (GCB) |
| 6 | ♣ | subdomain | Site: Single Cell Subdomain |
| 7 | ♣ | subdomain | Site: Human Cell Atlas Subdomain |
| 8 | ♣ | subdomain | Subdomains Combined |
| 9 | ⌾ | subdomain | FAQ: Use Our Single-Cell Omics Lab |
| 10 | 🖥 | sustainability | Seurat Version of Clustering PBMCs with Scanpy Tutorial (with new Seurat tool suite)[18] |
| 11 | 💡 | sustainability | Feature: Maintainer Page |
| 12 | ⌾ | sustainability | FAQ: How Do I Find the Maintainer Home Pages? |
| 13 | ℃ | sustainability | Single-Cell RNA-seq Preprocessing: 10× Genomics v3 to Seurat and Scanpy Compatible Format |
| 14 | ℃ | sustainability | Single-Cell RNA-seq Preprocessing: 10× Genomics CellPlex Multiplexed Samples |
| 15 | ℃ | sustainability | Single-Cell RNA-seq Analysis: Scanpy Preprocessing and Clustering |
| 16 | ♣ | engaging users | News: From GTN Intern to Tutorial Author to Bioinformatician |
| 17 | ♣ | engaging users | Course: Visual vs. Textual scRNA-seq |
| 18 | ⌾ | engaging users | FAQ: How can I talk with other users? |
| 19 | 🖥 | engaging users | Updating tool versions in a tutorial[19] |
| 20 | ♣ | engaging users | Event: SPOC CoFest |
| 21 | ♣ | engaging users | Event: SPOC CoFest - How did it go? |
| 22 | ♣ | engaging users | Event: SPOC HDR UK ELIXIR-UK CoFest 2025: How did it go? |
| 23 | ♣ | engaging users | News: Community Pages |
| 24 | 💡 | engaging users | Feature: Community Pages |
| 25 | 💡 | engaging users | News: Credit where it is due: GTN reviewers in the spotlight |
| 26 | 💡 | engaging users | News: GTN Helper Bots Connect the Galaxy |
| 27 | 💡 | engaging users | Feature: Topic-specific training digests for matrix channel |
| 28 | ♣ | engaging users | Event: SPOC Write-a-thon |

## Engaging users

SPOC champions collaboration between users and developers. Users are encouraged not only to take advantage of our materials but also to provide feedback or even become contributors themselves (Table 1, row 16).

### Using our resources

We have seen a rapid rise in users of our SPO tools since 2021, with over 5,000 different users. Over 300,000 jobs have been run on the current main public Galaxy servers (.eu, .org, .org, and .au) (Figures 2A–2C). In addition to quantitative metrics, we are also beginning to see the biological insights generated by Galaxy Users applying these tools. Applications range from malaria to cancer research to heart disease[40,41,42], even, at times, presenting back to the Galaxy community contributors themselves[43].

Our SPO training materials have consistently attracted around 1,000 views per month, with significant peaks during training events (Figure 2E). One such event was a week-long bioinformatics bootcamp held by the Open University (OU) (Table 1, row 17). This event introduced students to bioinfor-matics both in coding environments and via the Galaxy user interface.

### Contributing to our resources

SPOC aims to reduce barriers to user engagement. Galaxy developers typically converse on the Matrix chat forum, but we pushed for bridge formation between these chats and Slack, the platform where Galaxy communicates with users during training. We encourage users to get more involved by posting monthly welcome messages with a shareable snippet showing users how to engage with each other or join SPOC (Table 1, row 18).

SPOC has become a diverse community of users and developers with different skills, enabling us to spread the load of creating and maintaining our resources. We categorize the tasks based on expertise required, such as Galaxy tool development, which requires software development skills, or updating tutorial text, which can be done by users. We then built training materials on how to update training materials (Table 1, row 19), thereby encouraging users to become GTN contributors. We also link Galaxy training events with follow-up community collaboration

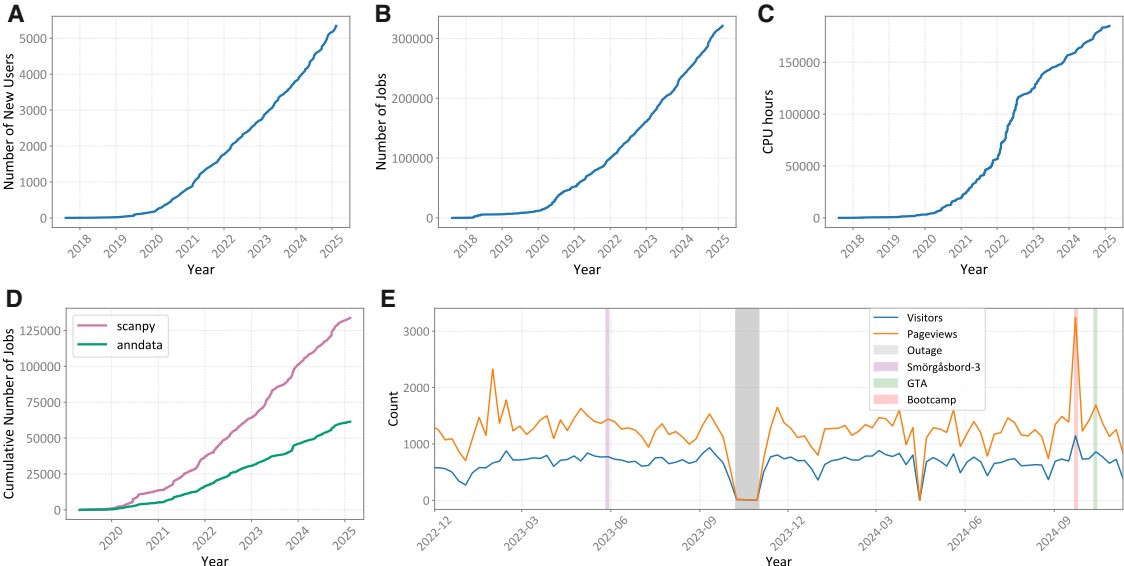

**Figure 2. SPO resources usage**

(A) New SPO Galaxy users (who have run at least one SPO tool), (B) number of jobs, and (C) computing resources for SPO tools over the period of 7 years.

(D) Cumulative number of jobs run using Scanpy and Anndata tools. Continuous updates and the incorporation of user feature requests have ensured their sustainability.

(E) Pageviews and visitors for GTN single-cell pages over time. This graph focuses solely on pages under single-cell topic to ensure data are not inadvertently collected for non-single-cell page visits (e.g., event pages with general audiences). Four annotations are marked, from left to right, they are as follows: GTN Smörgåsbord 3, which was an asynchronous learning event with many topics, including the large outage that occurred when Galaxy Europe administrators could not restore metrics gathering services, OU's bioinformatics bootcamp, which covered solely single-cell tutorials, and lastly, the recent Galaxy Training Academy (GTA) vent, which was similar in scope to GTN Smörgåsbord 3.

events. We piloted the first SPOC Collaboration Fest to onboard new members, with synchronous and asynchronous participation (Table 1, rows 20 and 21). The second SPOC Collaboration Fest ran during the Joint UK Hackathon from Health Data Research (HDR) UK and ELIXIR-UK (Table 1, row 22). To date, 70 scientists have contributed to SPO tools in Galaxy, and more than 30 scientists have contributed to SPO training materials (Figure 1C). Our expanding community of user-contributors ensures a user perspective in our materials.

### Acknowledging contribution

We pushed for the development of automated Community pages (Table 1, rows 23 and 24) to track and visualize our efforts, and for reviewers to be acknowledged in training materials (Table 1, row 25). Automated acknowledgments, including news posts for new tutorials in our developer and users channels, ensure all contributions are recognized and shared with the community (Table 1, rows 26 and 27). Finally, the writing of this manuscript was performed as a series of community events (Table 1, row 28).

## SPOC OUTCOMES AND RESOURCES

This section presents a consolidated list of outcomes, including Galaxy tools, workflows, training resources, and best practices for data management.

### New analyses

Here, we describe in depth the tools, workflows, training materials, and other resources developed within Galaxy to support

the growing needs of SPO analyses. It highlights important additions and enhancements beyond those described in our previously published single-cell portal.[15,16]

### Multiomics

While mRNA levels often correlate with protein levels, the complex regulation of transcription, translation, and the degradation rates of both mRNAs and proteins means this is not always the case.[44] Techniques that capture additional molecular layers at the single-cell level offer a more comprehensive view of cellular processes and serve as a valuable complement to transcriptomic analyses. We detail the next four key analyses available in Galaxy: cellular indexing of transcriptomes and epitope sequencing (CITE-seq), ATAC-seq, MUON (multimodal omics analysis), and single-cell Hi-C (scHi-C).

*Multiomics: CITE-seq.* CITE-seq combines single-cell transcriptomics with the use of oligonucleotide-tagged antibodies to quantify the expression of selected surface proteins.[45] CITE-seq data are supported in the latest Seurat tool suite within Galaxy (Table 2, row 1). Transcript and protein counts may be analyzed independently or integrated through the construction of a weighted nearest neighbor graph. This graph can then be used for clustering and further downstream analyses.

*Multiomics: ATAC-seq.* Single-cell ATAC sequencing (scATAC-seq) identifies open chromatin regions in individual cells, indicating which regulatory elements, such as promoters and enhancers, are accessible for transcription factor binding.[46] Galaxy incorporates several widely used scATAC-seq tool suites, including Sinto,[47] SnapATAC2,[48] and EpiScanpy[49] for preprocessing, clustering,

**Table 2. New analyses: Galaxy resources evolving with SPO technology**

| Item | Type | Subsection | Description |
|---|---|---|---|
| 1 | ✂ | multiomics | Seurat Version 5 |
| 2 | 🖥 | multiomics | ScATAC-seq Pre-processing[27] |
| 3 | ✂ | multiomics | EpiScanpy |
| 4 | ✂ | multiomics | Sinto |
| 5 | 🖥 | multiomics | Single-cell ATAC-seq Standard processing with SnapATAC2[28] |
| 6 | ✂ | multiomics | SnapATAC2 |
| 7 | 🖥 | multiomics | SnapATAC2 Batch Correction[29] |
| 8 | ✂ | multiomics | MUON |
| 9 | ✂ | multiomics | schicexplorer[30] |
| 10 | ✂ | spatial | Squidpy[31] |
| 11 | ✂ | spatial | SpaceXR |
| 12 | ✂ | spatial | spapros |
| 13 | ✂ | spatial | COSG |
| 14 | 🖥 | specialization | Analysis of Plant ScRNA-seq Data with Scanpy[32] |
| 15 | 🖥 | specialization | Removing the Effects of the Cell Cycle[33] |
| 16 | 🖥 | specialization | Scanpy Parameter Iterator[34] |
| 17 | ✂ | specialization | CELLxGENE |
| 18 | ✂ | specialization | CELLxGENE VIP |
| 19 | ✂ | specialization | decoupleR |
| 20 | ⌥ | specialization | Single-Cell Pseudobulk Differential Expression Analysis with edgeR |
| 21 | 🖥 | specialization | PseudoBulk Analysis with Decoupler and edgeR[35] |
| 22 | ✂ | specialization | InferCNV |
| 23 | 🖥 | specialization | GO enrichment Analysis on ScRNA-seq Data[36] |
| 24 | ⌥ | specialization | GO enrichment Analysis on ScRNA-seq Data Slides[37] |

and downstream analysis. The Sinto (Table 2, row 4) package facilitates barcode processing and demultiplexing. The scATAC-seq preprocessing tutorial (Table 2, row 2) explains how to map the reads from 10× ATAC-seq data, identify open regions of chromatin through peak calling, and generate an AnnData counts matrix using EpiScanpy (Table 2, row 3). The scATAC-seq analysis tutorial (Table 2, row 5) demonstrates how to cluster and annotate cells in an AnnData matrix using SnapATAC2 (Table 2, row 6) and Scanpy[50] Galaxy tool suites. Lastly, the multi-sample batch correction tutorial provides guidance on performing batch correction using the Harmony[51] and SnapATAC2 tools (Table 2, row 7).

*Multiomics: MUON.* The strength of multiomics is the ability to integrate data derived from different modalities in the same cell, offering a comprehensive understanding of cellular activity.[52] Built on top of Scanpy and Anndata, MUON is a multimodal omics analysis framework developed to integrate such multiomics single-cell data,[53] enabling researchers to explore complex cellular relationships and interactions. The Galaxy MUON tool suite (Table 2, row 8) facilitates the conversion between AnnData and MuData formats. It also supports normalization, clustering, integration with multiomics factor analysis, and data visualization.

*Multiomics: scHI-C.* scHi-C investigates the 3D chromatin structure at the individual cell level, providing insights into how promoters and enhancers interact on a per-cell basis. Additionally, by using a pseudobulk approach on clustered data, changes in chromatin structure throughout the cell cycle can be observed.[54,55] In Galaxy, the software scHiCExplorer was previously made available for demultiplexing, quality control, interaction matrix building, clustering, bulk matrix creation, and visualization (Table 2, row 9).[30]

*Spatial*

Spatial omics combines single-cell techniques with imaging to determine both the expression signatures and physical locations of cells within the tissue, enabling the study of the spatial relationships between cells and their surroundings.[56] Spatial transcriptomics can be categorized into two main approaches: imaging-based and sequencing-based technologies[57,58]. Imaging-based technologies use highly multiplexed single-molecule fluorescence *in situ* hybridization and advanced microscopy to detect the abundance and spatial location of RNA. Sequencing-based approaches, in comparison, use microarray chips to attach spatial barcodes to RNAs. Following barcode attachment, RNA is released and processed through standard next-generation sequencing protocols, inferring location from probe barcodes. We detail the next three key analyses available in Galaxy: Suidpy, SpaceXR, and Marker list generation. Developing training materials to accompany these tools is one of the future priorities for SPOC.

*Spatial: Squidpy.* Squidpy[59] is a Python framework that enables scalable analysis of spatially resolved omics data, which can be from either imaging- or sequencing-based approaches. Built on top of the popular single-cell analysis tool, Scanpy,

Squidpy provides advanced visualization tools to enable the user to easily map gene expression patterns onto spatially resolved cell types. In Galaxy, Squidpy was previously made available so that users can analyze their spatial data and visualize the results (Table 2, row 10).[31]

*Spatial: SpaceXR.* SpaceXR[60] is a framework for analyzing spatial transcriptomics data. It contains two computational methods: robust cell type decomposition (RCTD)[61] and cell type-specific inference of differential expression (CSIDE).[62] Sequencing-based spatial transcriptomics assays, which lack single-cell resolution (such as Visium), are limited in cell-type detection, as multiple cell types can be captured in one block. RCTD enables accurate cell type deconvolution in spatial transcriptomics data by mapping a corresponding single-cell reference data onto spatial transcriptomics data. CSIDE can additionally identify differentially expressed genes (DEGs) across spatial regions within specific cell types. We integrated SpaceXR functions into Galaxy (Table 2, row 11).

*Spatial: Marker lists.* Imaging-based methods require a suitable marker list since they can detect only a certain number of targets (e.g., genes) at the same time. The number of detectable targets in a single experiment has grown over the years, but remains limited in comparison to sequencing-based methods.[58] Therefore, a well-curated target list remains crucial for deriving meaningful insights. Our Galaxy tools integrate existing scanpy tools with functions from spapros[63] and COSG (COSine similarity-based marker Gene identification),[64] which identify and evaluate markers (Table 2, rows 12 and 13).

### Specialization

SPO analyses must be tailored to the specific characteristics of the dataset and the research question at hand. In the following, we describe some specializations that Galaxy SPO offers in more detail.

*Specialization: Single-Cell analysis for plant biology.* Our Galaxy tutorial offers an end-to-end pipeline adapted to the unique challenges of plant systems, enabling clustering, marker gene detection, and visualization tailored to plant-specific transcriptomes (Table 2, row 14).

*Specialization: Customization and exploratory analysis.* SPO analyses are iterative, requiring the evaluation of multiple parameter sets. High-quality decision-making can enhance statistical power to the same degree as a 4-fold increase in sample size.[65] We therefore developed training materials that guide users through key analytical decisions, namely the regression of confounding factors such as cell cycle effects (Table 2, row 15) and the simultaneous testing of multiple parameters (Table 2, row 16).

An important part of iterative data analysis is to visualize and explore results. CELLxGENE is a web-based, interactive, scalable tool that enables users to visualize their single-cell data, interactively select cells on the embedding, annotate them, and create gene lists of interest.[66] CELLxGENE-VIP extends this functionality by integrating customized visual analytic plots in high resolution. It also allows users to dive deeper into data, such as marker gene identification, differential gene expression, and enrichment analysis. Users can also visualize multiomics datasets and spatial transcriptomics with histological images.[14] In Galaxy, users can visualize Anndata objects with CELLxGENE

and CELLxGENE-VIP interactive tools (Table 2, rows 17 and 18) and export their annotations and gene sets.

*Specialization: Differential gene expression.* Frequently, users want to compare expression differences between experimental conditions or groups within clusters. This is statistically challenging because cells from the same individual are not independent observations, and failing to account for this inflates the risk of type I errors[67,68]. Pseudobulking addresses this issue by aggregating expression counts for each individual within a cluster. This approach enables statistically robust comparisons across groups and clusters using established bulk differential expression (DE) methods. In particular, the pseudobulk tool decoupleR[69] integrates with commonly used bulk RNA-seq DE tools such as edgeR.[70]

In Galaxy, we have integrated the decoupleR (Table 2, row 19) and the existing edgeR suite into a workflow for pseudobulk DE analysis (Table 2, row 20), and developed a tutorial (Table 2, row 21). The workflow generates pseudobulk counts from an AnnData object obtained through scRNA-seq analysis. DE analysis is then performed using edgeR, with integrated data sanitation steps to ensure robustness and minimize issues during execution. To facilitate interpretation, the workflow includes a volcano plot tool for intuitive visualization of DEGs. The resulting outputs are fully compatible with existing Galaxy tools from the transcriptomics community, enabling downstream gene set enrichment analyses or additional visualizations through volcano plots to explore DEG patterns.

*Specialization: Copy number variation.* The InferCNV[71] tool identifies somatic, large-scale chromosomal copy-number variations from scRNA-seq data. By comparing the expression intensity of genes in tumor samples to their intensity in control samples, InferCNV identifies which parts of the tumor genome are amplified or deleted. In Galaxy, users can analyze their expression matrices with the InferCNV tool (Table 2, row 22).

*Specialization: Functional analysis.* SPO analyses often generate long gene lists, whose biological relevance is not immediately apparent. Gene ontology (GO) analysis facilitates the interpretation of gene lists by classifying genes into three predefined functional categories: biological processes, molecular functions, and cellular components. The GO terms within each category are both machine-readable and human-interpretable, making them a powerful tool for deriving biological insights from complex datasets.[72] Tools such as GO enrichment and gProfiler[73] identify significantly enriched GO terms through statistical testing of a given gene list against a background gene set. In Galaxy, the pre-existing GOEnrichment and gProfiler tools have now been updated to allow for single-cell data and supplemented by training materials (Table 2, rows 23 and 24).

### RDM

Ensuring FAIR bioinformatics research is fundamental to the Galaxy community. SPOC members have worked with the ELIXIR Single Cell Omics community to help create the research data management (RDM) kit,[74] which sets out the best practices for single-cell RDM. We created a new subsection of tutorials for importing public data and converting between various data formats (Table 2, row 1). We aim to enable Galaxy users to manage SPO

**Table 3. Research data management: Galaxy resources integrating RDM with SPO analysis**

| Item | Type | Subsection | Description |
|------|------|-----------|-------------|
| 1 | ✤ | RDM | News: FAIR Data Management in Single-cell Analysis |
| 2 | 🖥 | data sharing | FAQ: Using Answer Key Histories |
| 3 | 🖥 | data sharing | FAQ: Input Histories & Answer Keys |
| 4 | 🖥 | data sharing | FAQ: Archive a History |
| 5 | 🖥 | data sharing | Single-cell Formats and Resources[76] |
| 6 | ✖ | data sharing | SCEasy Converter |
| 7 | ✖ | data sharing | AnnData to RI |
| 8 | 🖥 | data handling | Converting Between Common Single-cell Data Formats[77] |
| 9 | 🖥 | data handling | Matrix Exchange Format to ESet \| Creating a single-cell RNA-seq ref- erence dataset for deconvolution[78] |
| 10 | 🖥 | data handling | Bulk Matrix to ESet \| Creating the Bulk RNA-seq Dataset for Deconvolution[79] |
| 11 | 🖥 | data handling | Importing Files from Public Atlases[80] |
| 12 | 🖥 | data handling | Converting NCBI Data to the AnnData Format[81] |
| 13 | ⊘ | data handling | FAQ: Importing Data from Sierra LIMS |
| 14 | ✖ | data reuse | MuSiC |
| 15 | 🖥 | data reuse | bulk RNA deconvolution with MuSiC[82] |
| 16 | 🖥 | data reuse | MuSiC-deconvolution: compare[83] |
| 17 | 🖥 | data reuse | Evaluating Reference Data for Bulk RNA Deconvolution[84] |
| 18 | ⇌ | data reuse | Bioinformatics Projects: Using deconvolution to get new insights from old bulk RNA-seq data |

data effectively through embedding RDM principles in tools and tutorials.

### Data sharing
Galaxy enables data preservation and sharing in various forms, allowing users to easily comply with FAIR data principles.[9] We demonstrate the value of data sharing by using public data as inputs for many of our GTN tutorials.[75] Working with these imperfect, real-world datasets enhances users' data management skills while providing insights into how to make their own data easier to reuse.

We also piloted the use of answer key histories in GTN tutorials, allowing learners to import and reuse completed versions of the analysis for troubleshooting or checking their own results (Table 3, row 2). The GTN now includes answer keys in tutorial headers (Table 3, row 3). We also embedded instructions on history sharing in all introductory training materials, encouraging trainees to share histories to request help from trainers, to compare outputs when tutorials suggest alternate parameters,[75] and to develop the habit of data sharing early in their bioinformatics research.

Finally, we raised the need to enable history archiving within the wider Galaxy community, in order to preserve training and manuscript-associated data histories. With this new feature, we have archived all our training histories to preserve them for all future training, and established this as best practice for Galaxy contributors (Table 3, row 4).

### Data handling
The SPO field is rife with conflicting data types, formats, and varying levels of metadata in public repositories.[85,86] Handling these diverse datasets frustrates learners and inhibits data reuse. We built tools, workflows, and tutorials to help users manage the most common SPO data types and public repositories.

Users can start with a slide deck explaining the common data formats and structures (Table 3, row 5). Users can convert between common formats such as AnnData,[87] Single-CellExperiment,[88] SeuratObject[89] using SCEasy (Table 3, row 6) or anndata2ri (Table 3, row 7), which specifically converts between AnnData and SingleCellExperiment. SCEasy comes with many idiosyncrasies for declaring input and output formats that can be opaque even to proficient bioinformaticians. The Galaxy SCEasy tool coerces the correct parameters for the desired conversion invisibly for the user. The allowed conversions are not symmetric, however, these details are included in the tool to guide users (Figure 1H]).

We created a new GTN subtopic, "Changing data formats & preparing objects," which includes a tutorial on SPO data conversion using these tools (Table 3, row 8). One special case is the preparation of bulk and SPO datasets for deconvolution. We created resources to convert bulk RNA-seq samples and a scRNA-seq reference dataset into ExpressionSet format[90] for use with Multi-Subject Single Cell deconvolution (MuSiC)[91] toolsuite (Table 3, rows 9 and 10).

This GTN subtopic also covers the use of public data, including tutorials on importing datasets from the NCBI and Single Cell Expression Atlas[92] public repositories into Galaxy and preparing them for analysis (Table 3, rows 11 and 12). An FAQ was also added to help users import data from their institutional laboratory information management systems (LIMS) (Table 3, row 13).

### Data reuse
Deconvolution analysis estimates cell-type proportions in bulk RNA-seq data using single-cell reference datasets.[93] The MuSiC toolsuite leverages cross-subject and cross-condition information to improve deconvolution accuracy, enabling robust cell-type proportion estimates even when reference and bulk

datasets originate from different sources, as is commonly the case when using public data.[91]

We integrated the MuSiC suite into Galaxy (Table 3, row 14), supplemented by three tutorials (Table 3, rows 15–17). The first tutorial introduces MuSic and its use to explore a tissue. The next tutorial covers how to compare cell-type proportions across variables, such as disease status. Finally, a benchmarking tutorial enables users to evaluate how well their reference dataset performed and how accurate their deconvolution is likely to be. We combined these with existing training materials to develop a learning pathway, to onboard not only professional scientists but even undergraduates into reusing public data to gain new insights (Table 3, row 18).

## DISCUSSION

Our vision is that global scientists are empowered to perform FAIR SPO analysis following best practices in RDM, integrating both usability and quality in equal measure.

For usability, we focused on ensuring community engagement across regions and scientific fields to ensure the greatest use by the greatest number. We worked to unite across regional communities, ensuring that scientists in Australia and Europe can access and use the same materials and have an equal voice in resource development and innovation. We strive to actively build a culture engaging across disciplines in SPOC, from developers to trainers to users. This is most notably seen in our efforts to integrate different Galaxy sites—the Galaxy Community Hub—which is typically developer-focused; the Galaxy Training Network, which is aimed at newer Galaxy users; and the Galaxy interface itself, aimed at general users and now curated for SPO users across the globe (Figure 1G).

For RDM, the current state of our offering is summarized in (Figure 3). The data life cycle consists of six main stages: collect, process, analyze, preserve, share, and reuse.[94] Through a combination of SPO-specific materials, such as tools and workflows, and global Galaxy features, particularly on reproducibility and sharing, our comprehensive offering now covers all six stages. Prioritizing RDM principles in open-source communities can be difficult due to the lack of funding or acknowledgment for such activities,[95] which is why we have focused on automation and embedding principles in working practices, rather than aiming to significantly upskill the users.

### Ongoing challenges

Challenges remain in building, expanding, and sustaining open source analysis software on a global scale. User retention in documenting and following problems to solutions remains challenging. User churn[96] is directly related to the time it takes for an issue to be solved. The retention of "one-time contributors" is a common problem in open-source communities,[97] wherein first interactions—often on GitHub issue postings, for example—are a key indicator of user engagement, and again speed of integration impacts retention.[98] The demographics of user engagement are also a consideration for biasing future community development; it is well known that user demographics influence the likelihood of their engagement with a community, such as reporting software bugs,[99] leading to unequal representation and influence in the resulting software.

Maintenance of open-source software is also a common problem. While democratic, the lack of centralized support or ownership of open-source software often leads to abandonment and poor sustainability. One study found that 15% of GitHub repositories were unmaintained within a year,[100] while another found that the median life span of a software repository was 15 days.[101] The support of so-called "human infrastructure," most often through funding mechanisms, is a critical determiner of software maintenance[102,103]. The first single-cell training materials in Galaxy were launched in 2019, the inaugural papers in 2020 and 2021, and our community, SPOC, was formed in 2022. A commitment to acknowledgment and reciprocation of effort has enabled this long-lasting community. Nevertheless, continued and expanded support for not only software development but also software maintenance, alongside community engagement to ensure positive experiences for new members and increased likelihood of user retention and contribution, is key for the future of SPOC.

Tool selection and defining best practices are also challenges with the moving target of SPO technologies. Exchange of Galaxy tool development knowledge to build tool development capacity, as well as careful prioritization of the tools, is key. For the latter, a combination of community engagement with new and current users to define what they need, as well as with SPO expert analysts outside of Galaxy—i.e., Galaxy collaboration with non-Galaxy bioinformaticians—is a growing priority for SPOC. The first SPOC Collaboration Fest brought in more non-Galaxy users than Galaxy users, helping bridge this gap and ensure domain expertise in selecting the next SPO tools and features to bring into Galaxy.

The scale of SPO technologies is increasingly impacting the free-to-use public Galaxy infrastructure. While data archiving-to-Zenodo features enable the removal of large, unprocessed datasets after processing, the scale of data will only grow. The need for increased computing resources, albeit temporarily, for larger, atlas-level analysis, will soon become a problem.

Lastly, the globalization efforts that SPOC has spearheaded to unite disparate communities and build on centralized efforts to avoid work duplication remain challenging. How servers define and organize tool and tool version offerings remains disparate; engagement with the concept of subdomains and the usability of centralized resources is challenging; and creating collaboration across such distinct user groups globally, where discussions must be able to occur asynchronously to ensure diverse global representation, is challenging. This globalization challenge is not unique to SPOC; open-source software outputs generally remain clustered around Silicon Valley, London, and Berlin,[104] requiring further efforts to integrate and positively welcome new contributors from underrepresented regions.

### Future

Our next steps, as a community, will be to address these challenges. Our initial pilot of Collaboration Fests shows that if we welcome new contributors both within and outside of Galaxy, we can build platform-specific resources, review workflows,

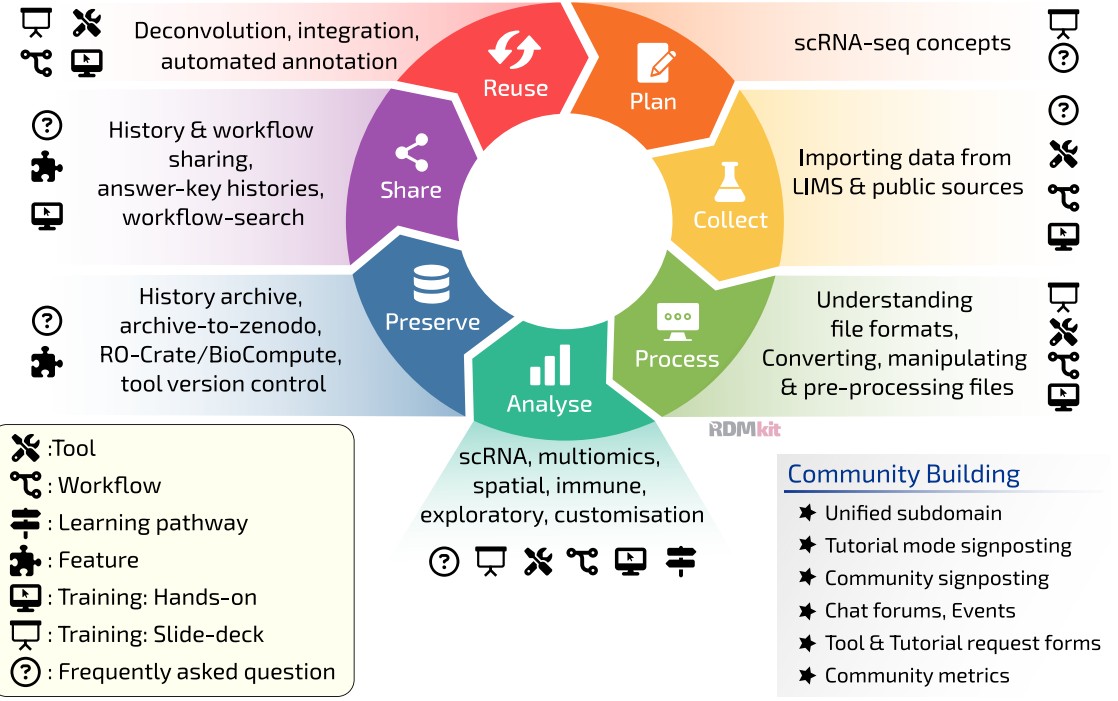

**Figure 3. Using the Galaxy SPO resources: We can plan experiments using conceptual slide decks that address concepts of batch correction and interpretation, along with exemplar paper replications**

We can collect data from LIMS as well as EBI and Human Cell Atlas single-cell data repositories using retrieval tools. We can process datasets, interconverting between formats using dedicated tools and workflows, from various protocols like Drop-seq, 10× Chromium, inDrop, and Smart-Seq. We can analyze data using our plethora of training materials and workflows. This includes standard clustering and annotation methods, pseudobulk analyses, trajectory analyses, and the integration of multiomics data. We preserve data—Galaxy allows containerization of data and metadata into research object standards like Research Object Crate and BioCompute objects. These formats bundle together analyzed data, metadata, and analysis parameters, creating a comprehensive package that can be easily shared and reused. We can share data and analyses effectively using Galaxy histories and the archive-to-Zenodo feature, as well as Pan-Galactic workflow search functions. Finally, we emphasize data reuse in Galaxy, providing specific tools, workflows, and training on how to reuse old data to create new insights—most notably in our bulk RNA deconvolution materials. Image modified from original[94]; ELIXIR (2021) research data management kit. A deliverable from the EU-funded ELIXIR-CONVERGE project (grant agreement 871075). URL: https://rdmkit.elixir-europe.org/.

and update conceptual training - leaving the tool development for tool developers. The sustainability and expansion of SPO in Galaxy are dependent on such community support. We must also stay up to date with emerging technologies—such as single-cell long-read analysis,[105] perturbation modeling,[106] and foundation models in SPO research[107]—while also fostering an active, collaborative, and inclusive community.

## Conclusion

The Galaxy SPOC offering has increased substantially in the last four years, driven by cross-disciplinary communication and global collaboration. We have more than 175 SPO tools and 120 training resources that enable cutting-edge analysis and address known user bottlenecks. We established community standards and infrastructure to enable collaboration and inform roadmap exercises. Since initial publications in 2020 and 2021, we have welcomed over 70,000 unique visits to our training materials and enabled the running of over 300,000 SPO jobs. SPOC has rapidly moved from a minor component of the transcriptomics training and tooling community to a trailblazer across the Galaxy ecosystem.

### ACKNOWLEDGMENTS

The authors extend their gratitude to the Galaxy community for supporting the testing of workflows as well as the development of tools. We thank Gareth Price for support in testing on the Australian Galaxy instance and training course participants for testing tutorials in live user settings. The authors acknowledge the support of the Freiburg Galaxy Team, University of Freiburg (Germany), funded by the German Federal Ministry of Education and Research BMBF grant 031 A538A de.NBI-RBC and the Ministry of Science, Research, and the Arts Baden-Württemberg (MWK) within the framework of LIBIS/de.NBI Freiburg. Finally, we acknowledge all contributors to SPO in Galaxy in the past, upon which many of the described materials have been built. We have aimed to highlight and cite those contributions throughout.

Internships were funded in part by the Engineering & Physical Sciences Research Council Training grant DTP (EP/T518165/1), as well as Hobart and William Smith Colleges (Geneva, NY, USA). Additionally, part of the materials was created thanks to the ELIXIR-UK: FAIR Data Stewardship Training (MR/V038966/1), as well as the Third Training Open Call issued by EOSC-Life, which has received funding from the European Union's Horizon 2020 program under grant agreement no. 824087. The development of tools and part of training resources was supported by the German Research Foundation Centres of Research Excellence SFB1425 (DFG #422681845), Baden-Württemberg Ministry of Science, Research and Arts (Ministerium für Wissenschaft, Forschung und Kunst), ELIXIR Germany/German Network for Bioinformatics

Infrastructure (de.NBI), and as a part of GHGA–The German Genome-Phenome Archive (www.ghga.de, grant no. 441914366 (NFDI 1/1)). The development of tools and workflows was partly supported by the PERSIST-SEQ project. The PERSIST-SEQ project has received funding from the Innovative Medicines Initiative 2-joint Undertaking under grant agreement no. 101007937. This joint undertaking receives support from the European Union's Horizon 2020 research and innovation program and EFPIA.

## AUTHOR CONTRIBUTIONS

Conceptualization, W.B.; data curation, W.B, M.H, and J.J.; funding acquisition, W.B., R.B., B.G., and P.V.; investigation, M.L, A.N.N., M.H., and J.J.; methodology, D.C.; project administration, W.B. and P.V.; software, W.B., M.L., A.N.N., F.H., P.M., P.V., M.H., B.G., and M.T.; software workflows, P.M., P.V., T.S., M.T., and M.G.; supervision, W.B. and P.V.; validation, W.B. and P.V.; visualization, M.H., P.V., and H.R.; writing – original draft, W.B., M.L., A.N.N., M.H., P.V., D.C.; writing – review and editing, F.H., D.C., J.J., H.R., S.H., and R.B.

## DECLARATION OF INTERESTS

The authors declare that they have no competing interests.

## DECLARATION OF GENERATIVE AI AND AI-ASSISTED TECHNOLOGIES IN THE WRITING PROCESS

During the preparation of this work, the authors, particularly for non-native English speakers, used ChatGPT and Copilot to refine sentences. After using this tool or service, the author(s) reviewed and edited the content extensively and take full responsibility for the content of the publication.

## SUPPLEMENTAL INFORMATION

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
