## [Document S1. Transparent peer review records for Marisa Loach et al · Cell Genomics]

Cell Genomics, Volume 5

Supplemental information

Galaxy single-cell & spatial omics

community update: Navigating

new frontiers in 2025

Marisa Loach, Amirhossein Naghsh Nilchi, Diana Chiang, Morgan Howells, Florian Heyl, Helena Rasche, Julia Jakiela, Mehmet Tekman, Menna Gamal, Pablo Moreno, Saskia Hiltmann, Timon Schlegel, Björn Grüning, Rolf Backofen, Pavankumar Videm, and Wendi Bacon

Galaxy Single-cell & sPatial Omics Community (SPOC) Update: Navigating new frontiers in 2025

Author list

Marisa Loach, Amirhossein Naghsh Nilchi, Diana Chiang, Morgan Howells, Florian Heyl, Helena Rasche, Julia Jakiela, Mehmet Tekman, Menna Gamal, Pablo Moreno, Saskia Hiltmann, Timon Schlegel, Björn Grüning, Rolf Backofen, Pavankumar Videm, and Wendi Bacon

Summary

Initial submission: Received : 23rd May 2025

Scientific editor: Judith Nicholson

First round of review: Number of reviewers: 2
Revision invited : July 15th 2025
Revision received : July 30th 2025

Second round of review: Number of reviewers: N/A
Accepted : 18th August 2025

Data freely available: Yes

Code freely available: Yes

This transparent peer review record is not systematically proofread, type-set, or edited. Special characters, formatting, and equations may fail to render properly. Standard procedural text within the editor's letters has been deleted for the sake of brevity, but all official correspondence specific to the manuscript has been preserved.

Referees' reports, first round of review

Reviewer 1

Here, Loach et al present an overview of the Galaxy SPOC. The manuscript gives a solid overview of how the platform has grown since 2022 and it now offers over 175 tools and 120 training materials. It does a great job highlighting the community's contributions, sustainable development efforts, and focus on FAIR data practices. An important section of the manuscript is where the authors highlight areas of concern and where improvements is needed. I found this to be honest self-reflection, and it provides a clear roadmap on how SPOC can be improved. Nevertheless, I believe that there are several major issues that need to be addressed prior to publication:

1. There are still some areas for improvement that were overseen by the authors. The fact that there was a need to "unify" different subdomains and improve navigation suggests that the overall structure is still a bit fragmented, which might make it harder for some users to find what they need and get started smoothly. Is there a strategy to improve this?
2. The manuscript includes several GitHub links, but it would be really helpful to also include direct links to the main Galaxy server and the Galaxy Training website. These are essential entry points for users and would make the manuscript more practical for readers interested in getting started.
3. While the manuscript emphasizes spatial analysis as a new feature, the Galaxy Training website does not have a clearly labeled section for it. Some relevant tutorials are buried under the "image" category, but that section mixes different types of imaging, and it is not obvious where to find tutorials for sequencing-based spatial analysis, which the manuscript says is supported.
4. It looks like spatial analysis tools are hosted on a specific Galaxy instance (<https://cancer.usegalaxy.org/>), but this is not clearly mentioned. Adding this info to the server homepage would help users know where to go.
5. Galaxy offers a lot of great tutorials, but they are not always easy to connect with the tools on the servers. For example, the Galaxy Cancer server lists many tools for multiplex tissue imaging (like Mesmer, ASHLAR, Cellpose), but there is no clear guidance on how to use them together in a full analysis pipeline.

Including links to relevant tutorials directly in the server would make things much easier for users.

6. Many workflows require specific input formats like .h5ad files, but there are no example datasets in the "Data Libraries." Providing sample inputs would help users test workflows and understand how to use them.

7. Some parts of the manuscript, especially the "spatial" section under "New Analysis," read more like technical documentation than a scientific paper. That section mostly introduces spatial technologies in general, rather than focusing on what Galaxy can actually do with them.

Reviewer 2:

The paper given an update perspective of the Galaxy web platform and its evolution into a platform for single-cell and spatial omics discovery. It gives an insight in the perpetual complexity of developing community software / web portals and making sure that these are regularly updated, maintained and extended. The paper here is a bundling of a number of community events, and nicely gives a reality check and an overview of all issues.

I only have minor comments:-

I miss a few key other platforms that envision the same: community data hosting and visualization tooling, being:

- Cell Annotation Platform (for community annotation)
- Vitesse (particular for spatial omics)
- ManiVault Studio (for low-latency interactive visual analyses)
- CIRRO (for spatial OMICS analysis and integration with other single cell data).

Some of these may be of interest to point to, as other platforms that aim at the same overarching goal of data sharing, hosting and visualization.

Second suggestion: all figures now are very much driven by user statistics and infrastructure design, and I miss a bit a cool use-case that showcases the discovery potential that is unlocked by Galaxy: can you give one example in which it is clear that Galaxy in a unique way "maximizes the information integration" for hypothesis generation and discovery.

Other than that, I find that the heroic and continuing efforts of the community definitely warrant publication of this perspective paper.

Authors' response to the first round of review

Point-to-point response CELL-GENOMICS-D-25-00538

To the editor:

From the editorial side, we would suggest a few points.

- For readability, and to better conform with the article format, it would be good if some of the main figures could be combined into single figures with multiple panels (eg/ figures 1-5 and 9-10).
 - After carefully revising the perspective article type guidelines, we thematically combined the figures into three figures. Figures 1, 2, 3, 5, 8, 9 and 10 together become new Figure 1. Figure 4, 6 and 7 grouped together into new Figure 2. The Figure 11 remained as it is and is now indexed as Figure 3.
- There are also a lot of subheadings and subsections, it would be helpful to consider if some of these work as paragraphs without a distinct subheading, or can be combined to ease readability.
 - We considered this carefully, and agreed that there is an element of a readability issue - particularly given reviewers themselves seemed to miss connections! To help this, we have added summary sentences in the overviews of the sections "We now detail three of these tools...", and include the subsection title in the paragraph, i.e. "Multiomics: ATAC-seq" instead of "ATAC-seq". We agree that this has significantly improved the readability of the paper, thank you for your comment!
- We also recommend that the Spoc hand is introduced at the beginning, but not used at every mention of the word. During the initial read for a non-specialist the emoji interrupted the flow for the reader at times.
 - Thanks for the recommendation! We understand the concern that the emoji might interrupt the reading. We removed all the SPOC hand emojis except the one in the summary and another one in the introduction.

To the reviewer #1:

Here, Loach et al present an overview of the Galaxy SPOC. The manuscript gives a solid overview of how the platform has grown since 2022 and it now offers over 175 tools and 120 training materials. It does a great job highlighting the community's contributions, sustainable development efforts, and focus on FAIR data practices. An important section of the manuscript is where the authors highlight areas of concern and where improvements is needed. I found this to be honest self-reflection, and it provides a clear roadmap on how SPOC can be improved. Nevertheless, I believe that there are several major issues that need to be addressed prior to publication:

- 1) There are still some areas for improvement that were overseen by the authors. The fact that there was a need to "unify" different subdomains and improve navigation suggests that the overall structure is still a bit fragmented, which might make it harder for some users to find what they need and get started smoothly. Is there a strategy to improve this?
 - a) Added this to the manuscript in the Galaxy Subdomain section: "Indeed, our community advocacy on the importance of centralizing and streamlining user access to the resources they need helped drive the uptake of the Galaxy Labs Engine "(<https://github.com/usegalaxy-au/galaxy-labs-engine>) for building centralized subdomains."
 - b) We agree that the structure was fragmented. However, this has been a key area where SPOC as a community has shined - we unified the subdomains to ensure there was a single location for users to onboard. We ensured users were a part of making that subdomain, and have driven this forward for other communities to follow, to ultimately end this fragmentation that used to occur.
- 2) The manuscript includes several GitHub links, but it would be really helpful to also include direct links to the main Galaxy server and the Galaxy Training website. These are essential entry points for users and would make the manuscript more practical for readers interested in getting started.
 - a) Thank you for this helpful suggestion! We had indeed overlooked the added practical value of providing direct links to our resources, and we appreciate you pointing this out. We added links to all the Galaxy servers and the Galaxy Training Network single-cell page to the RESOURCE AVAILABILITY section.
- 3) While the manuscript emphasizes spatial analysis as a new feature, the Galaxy Training website does not have a clearly labeled section for it. Some relevant tutorials are buried under the "image" category, but that section mixes different types of imaging, and it is not obvious where to find tutorials for sequencing-based spatial analysis, which the manuscript says is supported.
 - a) While we already have tools to support spatial analysis on Galaxy, there are indeed no Galaxy tutorials on end-to-end analysis. This is our priority to develop training materials and are currently being developed in Spatial2Galaxy project (<https://elixir-europe.org/how-we-work/scientific-programme/commissioned-services/science/cmr/spatial2>) as part of ELIXIR

Cellular and Molecular Research Programme. We have therefore added the following statement in the Spatial subsection - "Developing training materials to accompany these tools is one of the future priorities for SPOC".

- 4) It looks like spatial analysis tools are hosted on a specific Galaxy instance (<https://cancer.usegalaxy.org/>), but this is not clearly mentioned. Adding this info to the server homepage would help users know where to go.
 - a) The cancer.usegalaxy.org does not have a comprehensive listing of single-cell or spatial tools. It is also not available to user accounts from different servers. Conversely, the singlecell.usegalaxy.org subdomain is available across servers with a consistent tool listing and documentation. Importantly, "Spatial" is a subheading of tools within this subdomain. For this reason, we would still direct users to the singlecell.usegalaxy.org subdomain for wider SPO use, particularly given that spatial analyses often use or integrate with more standard single-cell tools.
- 5) Galaxy offers a lot of great tutorials, but they are not always easy to connect with the tools on the servers. For example, the Galaxy Cancer server lists many tools for multiplex tissue imaging (like Mesmer, ASHLAR, Cellpose), but there is no clear guidance on how to use them together in a full analysis pipeline. Including links to relevant tutorials directly in the server would make things much easier for users.
 - a) This comment is astute, and indeed exactly why we have created a centralised subdomain! This subdomain includes both links to tools, as well as the workflows to run them together, and - importantly - any training available using those tools. This is all hosted directly on the server. Additionally - and also crucially! - tools themselves now link directly to any training materials using them.

Tutorials

There is 1 tutorial available which uses this tool. These tutorials include training for the current version of the tool. View all tutorials referencing this tool.

Tutorials available in 1 category ▾

Help Forum

- 6) Many workflows require specific input formats like .h5ad files, but there are no example datasets in the "Data Libraries." Providing sample inputs would help users test workflows and understand how to use them.
 - a) As a community, we have moved away from the use of Data Libraries due to the difficulty of putting these datasets - and keeping them updated - across numerous servers. Therefore, instead we have moved towards using "Input" and "Answer Key" histories (as included in the Research Data Management table, Table 3, item 3). This enables users on any server to import Galaxy histories from wherever they are currently hosted. Importantly, this also makes updating tutorials and creating new ones more feasible for new contributors and bench scientists - they don't have to find a script/programmatic way of ensuring their datasets go into data libraries across the milieu of regional Galaxy servers.
- 7) Some parts of the manuscript, especially the "spatial" section under "New Analysis," read more like technical documentation than a scientific paper. That section mostly

introduces spatial technologies in general, rather than focusing on what Galaxy can actually do with them.

- a) Thank you for pointing this out - we consider this a readability issue, as also caught by the editor, because the subsequent sections after the 'Spatial' overview contain the actual technologies in Galaxy, and what they can do. We have now updated the subsections and headings to improve readability, and better connect this evidence with the overview.

To the reviewer #2:

The paper given an update perspective of the Galaxy web platform and its evolution into a platform for single-cell and spatial omics discovery. It gives an insight in the perpetual complexity of developing community software / web portals and making sure that these are regularly updated, maintained and extended. The paper here is a bundling of a number of community events, and nicely gives a reality check and an overview of all issues.

I only have minor comments:-

- 1) I miss a few key other platforms that envision the same: community data hosting and visualization tooling, being:
 - Cell Annotation Platform (for community annotation)
 - Vitessce (particular for spatial omics)
 - ManiVault Studio (for low-latency interactive visual analyses)
 - CIRRO (for spatial OMICS analysis and integration with other single cell data).

Some of these may be of interest to point to, as other platforms that aim at the same overarching goal of data sharing, hosting and visualization.

- a) We appreciate the reviewer's suggestions. We included citable platforms from the reviewer's suggestions, such as Vitessce, as well as a few other platforms like Nextflow and CellxGene that we found particularly relevant. We extended the introductory paragraph of the "Galaxy for single-cell analysis" section with appropriate text.
- 2) Second suggestion: all figures now are very much driven by user statistics and infrastructure design, and I miss a bit a cool use-case that showcases the discovery potential that is unlocked by Galaxy: can you give one example in which it is clear that Galaxy in a unique way "maximizes the information integration" for hypothesis generation and discovery.

Other than that, I find that the heroic and continuing efforts of the community definitely warrant publication of this perspective paper.

- a) We thank the reviewer for this insightful suggestion! Although the primary focus of our paper is not to discuss specific case studies, we agree that including examples of how SPO resources have been used adds value. We have therefore incorporated a few in "Using our resources" subsection. We also appreciate the reviewer's kind and encouraging feedback.

Referees' report, second round of review

Authors' response to the second round of review